# A Novel Guided Zygomatic and Pterygoid Implant Surgery System: A Human Cadaver Study on Accuracy

**DOI:** 10.3390/ijerph18116142

**Published:** 2021-06-07

**Authors:** Francesco Grecchi, Luigi V. Stefanelli, Fabrizio Grivetto, Emma Grecchi, Rami Siev, Ziv Mazor, Massimo Del Fabbro, Nicola Pranno, Alessio Franchina, Vittorio Di Lucia, Francesca De Angelis, Funda Goker

**Affiliations:** 1IRCCS (Istituto di Ricovero e Cura a Carattere Scientifico) Orthopedic Institute Galeazzi, Via Riccardo Galeazzi, 4, 20161 Milano, Italy; dr.grecchi@tiscali.it (F.G.); massimo.delfabbro@unimi.it (M.D.F.); 2Private Practice, Viale Leonardo Da Vinci 256, 00145 Rome, Italy; gigistef@libero.it (L.V.S.); vittorio.dilucia@gmail.com (V.D.L.); 3Maggiore della Carità di Novara, Università degli studi del Piemonte Orientale “Amedeo Avogadro”, Reparto di Chirurgia Maxillo-Facciale, C.so Mazzini 18, 28100 Novara, Italy; grivettof@gmail.com; 4Private Practice, Via G. Boccaccio 34, 20123 Milano, Italy; emma.grecchi@gmail.com; 5Private Practice, Tel Aviv Basel St 46, Tel Aviv 6100000, Israel; ramisiev@gmail.com (R.S.); drmazor1@gmail.com (Z.M.); 6Department of Biomedical, Surgical and Dental Sciences, Università degli Studi di Milano, 20122 Milano, Italy; funda.goker@unimi.it; 7Department of Oral and Maxillo Facial Science, Sapienza University of Rome, Via Caserta 6, 00161 Rome, Italy; francesca.deangelis@uniroma1.it; 8Private Practice, Via Legione Gallieno 44, 36100 Vicenza, Italy; alessiofranchina@icloud.com

**Keywords:** zygomatic implant, guided surgery, computer aided implantology, navigation, dynamic navigation, surgical guides, surgical templates, pterygoid implants, CAD/CAM, accuracy, guidance

## Abstract

The aim of this human cadaver study was to assess the accuracy of zygomatic/pterygoid implant placement using custom-made bone-supported laser sintered titanium templates. For this purpose, pre-surgical planning was done on computed tomography scans of each cadaver. Surgical guides were printed using direct metal laser sintering technology. Four zygomatic and two pterygoid implants were inserted in each case using the guided protocol and related tools. Post-operative computed tomography (CT) scans were obtained to evaluate deviations between the planned and inserted implants. Accuracy was measured by overlaying the real position in the post-operative CT on the virtual presurgical placement of the implant in a CT image. Descriptive and bivariate analyses of the data were performed. As a result, a total of 40 zygomatic and 20 pterygoid implants were inserted in 10 cadavers. The mean deviations between the planned and the placed zygomatic and pterygoid implants were respectively (mean ± SD): 1.69° ± 1.12° and 4.15° ± 3.53° for angular deviation. Linear distance deviations: 0.93 mm ± 1.23 mm and 1.35 mm ± 1.45 mm at platform depth, 1.35 mm ± 0.78 mm and 1.81 mm ± 1.47 mm at apical plane, 1.07 mm ± 1.47 mm and 1.22 mm ± 1.44 mm for apical depth. In conclusion, the surgical guide system showed accuracy for all the variables studied and allowed acceptable and accurate implant placement regardless of the case complexity.

## 1. Introduction

The goal of every surgical procedure, including implantology, is to achieve the planned result after carefully evaluating the cost-benefit ratio. Many variables can be influential on the design of the project and the accuracy of the outcomes. The accuracy of the diagnostic phase, the quality of the materials used, the operator’s skills and expertise are all essential factors, and together with the advances in technology and the progressive improvement in the development of the devices used, allow one to achieve optimal clinical outcomes [1,2,3,4]. Currently, there is a continuous evolution and improvement in order to overcome operators’ limitations and to minimize the gap of precision placement between expert operators and professionals with less experience in advanced surgical techniques [5,6,7,8,9,10]. In the medical field, one of the first examples dates back to the late 1970s and is the well-known advent of the Russian mechanical staplers for gastrointestinal surgery, which allowed unexperienced surgeons in peripheral hospitals to the achieve results as excellent as experienced operators [11].

Currently, dynamic/static guided surgery is one of the hottest research topics in the field of conventional, pterygoid and zygomatic implantology [5,12,13,14,15,16]. The static guided surgery systems utilize surgical templates to guide the drilling process [15]. Dynamic navigation options plan and calibrate the ideal position of the implants by optical reference markers placed over the patient, and insert implants in accordance with the three-dimensional (3-D) image on navigation system using surgical instruments by means of a tracking system array [15,17,18,19]. Both of these guided surgery navigation methods for conventional dental implant placement have been widely evaluated and reported in literature with high accuracy levels as results [5,20,21,22].

In the guided implant placement, a pre-operative virtual plan and an accurate surgical diagnosis are crucial to evaluate the anatomical structures, in order to minimize the intra/post-operative complications and to improve the treatment outcomes [5,23,24,25,26,27]. Today, with the help of technological developments, it is possible to assess the 3-D anatomy of the patients and pre-operatively plan the ideal position of the implants, using the data provided by Computed Tomography (CT) and adequate surgical software programs [5,12,13,14,15].

The development of the imaging technologies known as Cone Beam Computed Tomography (CBCT) has led to a significant improvement in the pre-surgical planning, since it provides three-dimensional (3-D) data of the patient’s anatomy with less radiologic dose than Computed Tomography (CT). In addition, it is possible today to virtually place the dental implants in their ideal position, through various software programs, using the DICOM (Digital Imaging and Communication in Medicine) data provided by CT scans [5,23,24,25,26,27,28,29,30].

Computer-guided implant placement represents several advantages when compared to free-hand surgery, including minimally invasive surgery with a reduction of operative time and steps. Additionally, these protocols allow prosthetic-driven implant placement with more accurate results and simplified procedures, making them applicable even by less experienced clinicians [6,27,31,32,33]. Currently, the majority of the reports in literature involve studies with high level experienced operators. There are only a limited number of model-based studies investigating whether the surgical experience has an impact on implant placement accuracy while using drilling guides [34,35]. According to the literature, there is an improvement in the precision of computer-guided implant placements compared to conventional ones, however the reports evaluating the results between inexperienced and skilled surgeons are not consistent, although similar values of errors were found [14,35,36].

Zygomatic and pterygoid implants were suggested as an alternative treatment to massive grafting surgery in the severe atrophic maxillary. The typical zygomatic implant length, ranging from 35 to 60 mm, and the proximity to many anatomic limitations such as vessels, nerves and structures such as the orbit, makes this procedure a challenging one and exposes the operators to higher risks when compared with conventional dental implantology [37]. Stella and Warner in 2000 described the sinus slot technique to prepare the site between the base of the zygoma to the bone crest, avoiding injuries to the sinus membrane. This approach also helped to respect the ideal three-dimensional zygomatic implant site preparation as the following drills can work free from any deviation generated by the bone crest remnants [38]. One of the main problems with guided zygomatic implant insertion is the application of the methods deriving from traditional implantology (which is based on a two-dimensional view of the problem, to zygomatic implants, whose vision must be strictly kept in mind in the third angular dimension) [39,40]. A dedicated system for zygomatic implant placement based on a bone-supported surgical template seems to be reasonable to increase the safety and the accuracy. It is still difficult to achieve the correct driven angle of zygomatic osteotomies, and additional researches with randomized clinical trials are needed to assess the predictability of these procedures [13,40,41].

The aim of this cadaver study was to analyze zygomatic and pterygoid implant deviations when applying a novel surgical guide protocol for ZI/PI surgery, as an alternative to free hand placement. The accuracy was evaluated by merging the pre-operative and post-operative CT scan datasets to assess the effect of this novel surgical guide on implant deviations.

## 2. Materials and Methods

This study evaluates accuracy of zygomatic and pterygoid implant insertions, during a practical training on human cadavers with unexperienced surgeons (in zygomatic implant insertions). A total of 4 zygomatic and 2 pterygoid implants were placed in each cadaver’s head (10 cadaver heads in total), using DMLS (Direct Metal Laser Sintering) 3D printed titanium surgical templates.

The cadavers were donated by individuals for their use in scientific purposes and an official laboratory permission to work on cadavers was obtained from Italian competent authority (Prot. Nr 08-05 Maggio 2021). Common rules/guidelines applied in European Union which was used in this study while working on cadavers were as follows:Cadavers were treated with respect at all timesA professional attitude was applied during all lab proceduresHuman cadaver material was not removed from the laboratory under any circumstances.No photographs or video cameras were used in the laboratoryOnly health professionals enrolled in the course and instructors entered the labAll cadaver material remained at the assigned dissection tableIncomplete dissections or intentional destruction of dissected structures was considered unprofessional behavior and work area was kept as clean as possible.

The guide design was performed by a clinical plan based on the CT scan of each maxilla. The CT scan Gantry tilt was 0° and slices thickness were 0.4 mm. After implant insertions, a new CT scan was carried out to compare deviations between planned and achieved implants. Accuracy was measured by overlaying the real implant position in the postoperative CT on the virtual presurgical placement of the implants in the pre-operative CT scan. The accuracy evaluation involved angular and linear (coronal, apical and depth) deviations.

### 2.1. Presurgical Procedure

In brief, a pre-operative CT scan was taken for each cadaver and the resulting DICOM files were segmented, forming STL (Standard Triangulation Language) files. Using a dedicated planning software, both zygomatic (ZI) and pterygoid implants (PI) were planned (Figure 1) and the surgical templates were designed (Figure 2 and Figure 3). Each STL file of the maxillary bone with planned zygomatic and pterygoid implants became the baseline for the post-operative comparison. A post-operative CT scan of each cadaver’s head with implants was taken after the surgery.

The DICOM images of the post-operative CT were uploaded in a dedicated software (mimics Medical 19.0, Materialise Dental, Leuven, Belgium). Segmentation based on tissue density was carried out in order to separate implants from the surrounding bone.

The STL files of the maxillary bone with the planned implants, which were obtained from the first CT scan, were uploaded into the software. The superimposition of the pre-op and post-op CT images was achieved by using the best fit alignment tool (Figure 4 and Figure 5).

The planned and inserted implants were considered as cones with a base and an apex and their spatial coordinates (the center of the base and the apex) were registered by using a dedicated software (3-matic Medical 11.0, Materialise Dental, Leuven, Belgium) and were exported in an excel sheet in order to calculate coronal, apical, depth and angular deviations.

A diagnostic CT scan was performed to evaluate the residual maxillary bone anatomy in order to determine the location of ZI/PI sites using a 3D planning software. The implants’ angulations, positions, and dimensions as well as the inclinations of the multi-unit-abutments (MUA) were carried out using a dedicated implant surgical software (EZplan Real Guide, NORİS medical).

The ZIs were planned with an extra-sinus path with a lateral upward angulation of 45–60 degrees from the vertical axis. The implant’s apex was positioned to pass through the zygomatic bone in a bi-cortical manner in order to obtain the maximum anchorage. PIs entry points were designed to be 10–12 mm posterior to the tuberosity and the angulation was adjusted to join the pterygoid medial plate.

Once the surgical plan was defined, the data set allowed to design a CT-derived bone supported surgical guide with a novel layout and showed an optimal stability. To do that, the designed guide was exported as a STL (standard triangulation language) file to be fabricated using 3D printing processes.

#### 2.1.1. EZgoma Principle

The EZgoma guide is an apparatus for the placement of zygomatic implants previously planned by a dedicated software. The guide provided two separate supports for two ZIs on one side (Figure 6). Each support had a cylindric form divided into two parts (upper support on the buccal part and lower support on the palatal part). The lower support worked with the upper support creating an efficient system to avoid the bending momentum due to the rotational movement of the bur during drilling, which also allowed the alignment of the burs to the cylindric body.

#### 2.1.2. EZgoma Procedure

A palatal incision was carried out in the maxillary soft tissues with bilateral vertical posterior releasing incisions. The muco-periosteal flap was elevated to expose the alveolar crest, the piriform aperture, the lateral wall of the maxillary sinus, the infraorbital nerve emergence, the tuber maxilla, the central and the posterior part of the zygomatic complex (Figure 7).

The bone-supported surgical drill guide was placed and fixed with three 1.6 mm diameter mono-cortical osteosynthesis screws. These screws provided a stable fitting of the guide to the bone, preventing any tilting, which is crucial for the success of the guided surgery (Figure 8).

#### 2.1.3. Pterygoid Implant Protocol

When pterygoid implants are planned in addition to zygomatic ones, it is recommended to start the procedure with pterygoid implants placement in order to use the implants as anchor pin, in addition to the screw fixation.

The pterygoid osteotomy was performed by a long 2.8 mm diameter drill, used with a reduction spoon placed in a long sleeve defining the planned drilling direction. The marks on the drills were used to check the drilling depth (Figure 9).

The implant was seated by a driver through the guide (Figure 10) until the driver stopped on the sleeve. The planned orientation of the implant was achieved by aligning the hex of the driver with the hex of the sleeve (Figure 11).

#### 2.1.4. Zygomatic Protocol

After the surgical guide was fixed, the implant site preparation continued with a spherical diamond bur (∅4.2 mm) to create a notch in the bone (Figure 12) facilitating the bone approach of the next cylindric diamond bur (Figure 13).

The cylindric diamond bur was used to create a cylindric groove in the lateral wall of the maxillary sinus to enable the drilling tools to complete the osteotomy and to provide adequate bone support to the zygomatic implant. The cylindric diamond bur’s tip was placed between the bone and the upper support of the guide, that worked as a fulcrum for the medial movement of the bur against the sinus lateral wall, that was grinded until the bur was seated on the lower support. (Figure 13).

A 4.2 mm diameter drill was positioned between the guide supports and driven inwards up to a mark on the drill (Figure 14), removing the remaining bone under the upper support to allow a free setting of the following centering spoon.

The centering spoon (Figure 15) was placed in order to allow the bone site preparation with a 3 mm internal diameter. A centric drilling is always suggested in order to respect the original planning and to avoid a final implant deviation higher than usual. The drilling depth was determined once the drill was stopped by the spoon sleeve (Figure 16).

The drill No. 1 was used to finalize the bone preparation, taking care to align the bur with the upper and the lower support (Figure 17). The drilling depth was determined by aligning the planned depth mark on the drill with a reference slot on the guide (Figure 18) (The N. 2 and N. 3 final drills are used only in case of D1 bone).

A depth probe was inserted into the osteotomy through the guide and the depth of the osteotomy was assessed aligning the planned line on the probe with the mark on the guide (Figure 19).

The planned zygomatic implant was screwed into the osteotomy site through the opposite half-sleeve of the guide (Figure 20).

An implant driver was used to perform the implant’s seating until its final vertical position was aligned with the mark on the guide and the head geometry was helpful to control its final alignment. Moreover, a pin was also used to definitely orientate the prosthetic connection as planned, in order to respect the following correct placement of the selected angulated abutment (Figure 21).

Finally, the surgical guide was removed simply unscrewing the two fixation screws.

The above-mentioned guided approach allowed the placement of the multi-unit-abutment on the implants before the surgical guide was removed (Figure 22 and Figure 23).

## 3. Results

A database was created using Excel (Microsoft, Redmond, WA, USA). Data were evaluated using standard statistical analysis software (version 20.0, Statistical Package for the Social Sciences, IBM Corporation, Armonk, NY, USA).

Descriptive statistics including minimum and maximum values and mean ± SD values were calculated for each variable, and box plots were used to evaluate data outliers. The Shapiro–Wilk test was used to determine whether or not the data conformed to a normal distribution.

The independent-samples t-test was used to identify statistically significant differences in the accuracy of zygomatic implants compared to pterygoid implants and to evaluate differences in the intragroup analysis between implants positioned on the right and the left sides of the maxilla.

In each test, the cut-off for statistical significance was *p* ≤ 0.05.

In total, 10 cadavers were used for the study. In each cadaver heads were inserted four zygomatic implants, two for the left and two for the right side, and two pterygoid implants, one for each side (for a total of 40 zygomatic and 20 pterygoid implants).

The mean differences in the platform plane, platform depth, apical plane, apical depth and angle position between the zygomatic and pterygoid implants compared to the virtual implant planning are reported in Table 1 and Figure 24.

The independent-samples *t*-test showed no statistically significant mean difference in the platform plane, platform depth, apical plane, apical depth and a significant mean difference in the angle (*p* = 0.006) of zygomatic versus pterygoid implants (Table 1).

No difference was found in the accuracy between the left and right side in both zygomatic and pterygoid implants (Table 2).

## 4. Discussion

In order to prevent possible complications of implantology, numerous authors have proposed surgical navigation systems with the support of techniques aimed at increasing precision and decreasing the risks. In this study, an innovative system used for a safe placement of zygomatic and pterygoid implants is reported. Each implant was carefully planned, starting from a virtual plan, based on a 3-D CT-scan, using a specific software. The surgeries were carried out by innovative customized 3-D printed surgical guides and a dedicated surgical kit.

The free hand extra-sinus drilling protocol for zygomatic implants requires a three-dimensional visualization of the anatomy. The first step is to define the position of the multi-unit-abutment that have to be attached to the zygomatic implant on the alveolar ridge. Tracking a line connecting the entry point of the implant at the bone crest level with the zygomatic bone, it is possible to define the path of the bone preparation. Zygomatic and pterygoid implant insertions can be affected by many risks occurring during their planning and performing. Clear setting of the entrance point, trajectory path, and exit point of the implants, combined with a successful transition from the implant planning to the surgical phase, are all crucial factors [17,18,40].

This study tested a bone-supported technique that consists of a single sintered titanium template, placed during all the surgical procedures and clinically validated by the data emerging from the overlapping of the pre-operative planning with the post-operative CT of the specimens’ heads (Table 1 and Table 2).

The success of a guided procedure mostly depends on the precise position of the guide on the hard or soft tissues. Particularly, in cases of severe atrophic maxilla, it might be quite difficult to maintain the stability of the surgical guide throughout the whole drilling procedure [42]. A screw-retained surgical guide, fabricated with CAD-CAM (computer-aided design and computer aided manufacturing) technology, seems to make it feasible to ensure the accuracy and the safety of the final results [36,37,39,40]. The guide thus constructed was placed on the anterolateral wall of the maxilla and fixed to the bone surface by means of 3 screws with a diameter of 1.6 mm and was removed after implants and multi-unit-abutments were correctly placed. No additional nor more aggressive procedures were needed in terms of surgical access to the maxillary sinus.

A surgical guide for the placement of zygomatic implants fabricated in the same manner as conventional dental implants is considered less reliable, as these implants are significantly longer (35–60 mm) compared to conventional dental implants. Due to this fact, a slight error in the drill path direction and in the angular deviation can significantly alter the trajectory, the positions of the apex and the divergence at the exit point. In the event of deviations in zygomatic implant placement, the consequences can be much more serious than the complications of conventional implantology [37,41,42].

The use of the bone tissue as a supporting base has been considered mandatory, as well as the use of a rigid structural material as titanium, for the guide manufacturing, as both make it feasible to transfer the plan with absolute precision to the implant site. Moreover, due to the path of these extra-sinus long fixtures, a mucosal-supported guide cannot be feasible.

The guided templates for conventional implants, even the most advanced, provide occlusal sleeves to guide burs during osteotomy. The length of the above-mentioned sleeves usually ranges from 4 to 6 mm and they are suitable for implants within 15 mm. Besides, a 35 mm to 60 mm zygomatic sleeve would be exposed to the consistent risk that the bur may get stuck and may reduce handling.

The EZgoma inverted support system overcomes these difficulties by reducing the overall dimensions of the device, as it is based on a single bone-supported template consisting of two open, opposite half-sleeves, connected by a double track, in which it is possible to house the drills with extreme precision with a standard handpiece for low speed implantology. This is unconventional if compared with free hand zygomatic implants placement, which has usually been proposed to be performed by using a straight head handpiece.

The suggested surgical guide design made the entire guided surgery, as well as implant and abutment placement according to the planned project to support an immediate loading prosthesis, easier. In order to test implant deviations, the planned zygomatic and pterygoid implants have been saved as SLT files and compared with the ones obtained from the post-operative CT-scan. As shown in Table 1, both zygomatic and pterygoid implant deviations resulted in values comparable with those published for conventionally guided implants and no statistically significant differences have been reported [39,43,44].

This present cadaver study was performed in an anatomical laboratory environment, however in real life, clinicians perform zygomatic implant surgery on patients with extremely atrophic maxillary bone. Management of oral rehabilitation in such patients can be quite demanding, and one of the key factors is the careful follow-up period. The marginal bone resorption and changes in bone levels must be evaluated at least every two years. Especially for the specific extra-sinus placement that is typical of the presented surgical protocol, peri-implant mucosal situation must be additionally controlled. For this purpose, it is mandatory to annually remove the prostheses to check the oral hygiene status of prosthesis and the status of the abutments that are placed over the zygomatic implants. The radiographic assessment of the on-going peri-implant vertical bone loss after implant placement is considered as an essential issue for clinicians. Cosola et al., proposed a method of standardization of two-dimensional radiographs that can allow the clinicians to minimize the patient’s exposure to ionizing radiations for the measurement of marginal bone levels around dental implants [45]. Such methods can be critical in order to evaluate the bone changes around the zygomatic implants at the follow-up period and have a great impact on the long-term successful results.

In the present work, as the implants have been planned both on the right and the left side of the involved heads, a comparative analysis of the side-related deviations has been carried out. No statistically significant differences have been observed in terms of accuracy between the left and right side either in zygomatic or pterygoid implants. Since all the implants have been placed by the unexperienced surgeons involved in the clinical training on cadaver heads, the accuracy results gave evidence of the safety of this guided procedure.

In cases of guided implant surgery, patients with limited arch space can be a challenging situation, especially for zygomatic implant insertions. In such cases, various protocols were introduced in literature as a solution. De Santis et al., in a clinical study evaluated a novel radiologic protocol and a new occlusal radiographic index that can give the clinician the possibility of identifying patients with limited inter-arch space. As a result, the new radiological occlusal index made with condensation silicone (Sandwich Index) proved to be effective in reproducing the maxillary forced maximum opening position during the initial planning phase. Additionally, their method prevented errors in the inclusion or exclusion of patients suitable for NobelGuide treatment [46]. The EZgoma guide system represented in this study, is a suitable method even for patients with limited mouth opening. This guide system is fixed unilaterally, which is easy to use.

The proposed EZgoma method takes several advantages of conventional sleeve guides. The two opposite supports of the guide (Figure 1), the coronal one located palatal to the alveolar ridge and the apical one placed buccally, at the entry point of the zygomatic bone and on the lateral maxillary wall of the sinus, make easier the entire surgical procedure as they leave a certain degree of freedom to the surgeon to prepare the implant site. Moreover, there is a prosthetic advantage because of the extra-sinus implant placement, as it allows a better and natural emergence profile of the future prosthesis, avoiding an uncomfortable larger palatal volume.

## 5. Conclusions

Zygomatic and pterygoid implants have been suggested as a solution to rehabilitate severely resorbed maxillary bone. The proposed guided zygomatic and pterygoid surgery seems to be an easier and safer method when compared with the free hand approach. Further research on the accuracy of the entire procedure is mandatory in order to avoid critical surgical complications, which can involve accidents to the surrounding anatomy. This research, which utilized a new surgical guide design, in terms of accuracy between planned and placed zygomatic and pterygoid implants, resulted in very small deviations, comparable with the ones obtained with conventional surgical guides.

Zygomatic and pterygoid implant insertions represent an effective, quicker and less invasive treatment method in indicated cases, as compared to massive bone augmentation. According to the results of this study, in terms of accuracy and with respect to the pre-surgical planning, the procedure is feasible with successful results even if performed by unexperienced surgeons. However, the simplification of the surgery and the reduction of the invasiveness should be improved.

One of the limitations of this study is the fact that it is a cadaver study.

## Figures and Tables

**Figure 1 ijerph-18-06142-f001:**
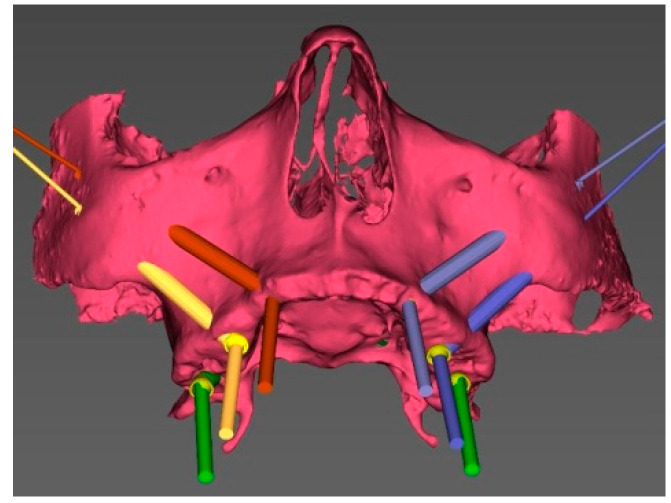
3D view of the pre-surgical plan for position of the zygomatic and pterygoid implants.

**Figure 2 ijerph-18-06142-f002:**
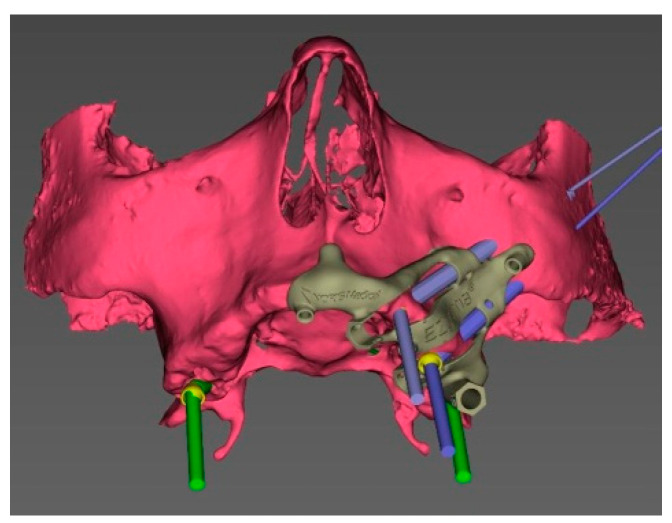
3D view of the pre-surgical plan with the surgical guide on the left side.

**Figure 3 ijerph-18-06142-f003:**
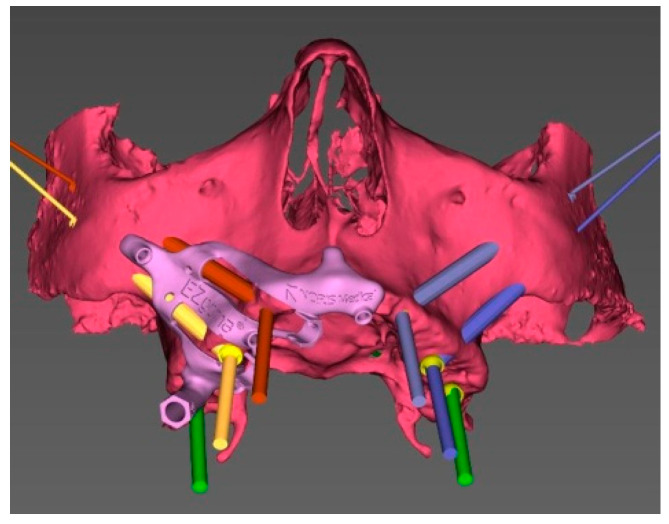
3D view of the pre-surgical plan with the surgical guide on the right side.

**Figure 4 ijerph-18-06142-f004:**
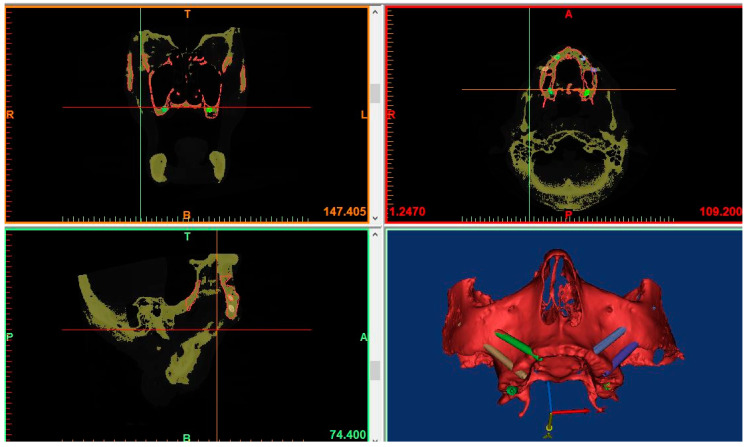
Pre-operative and post-operative CT-scans superimposed and aligned for evaluation of planned and placed implants.

**Figure 5 ijerph-18-06142-f005:**
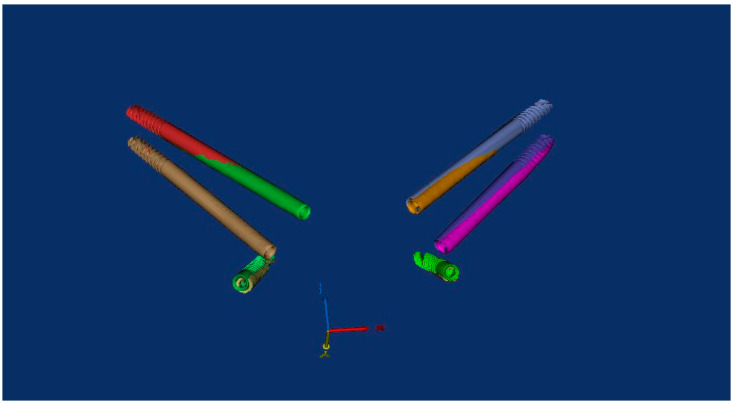
Superimposition of the pre-surgical plan and the post-operative placement of the implants showing minimal deviations.

**Figure 6 ijerph-18-06142-f006:**
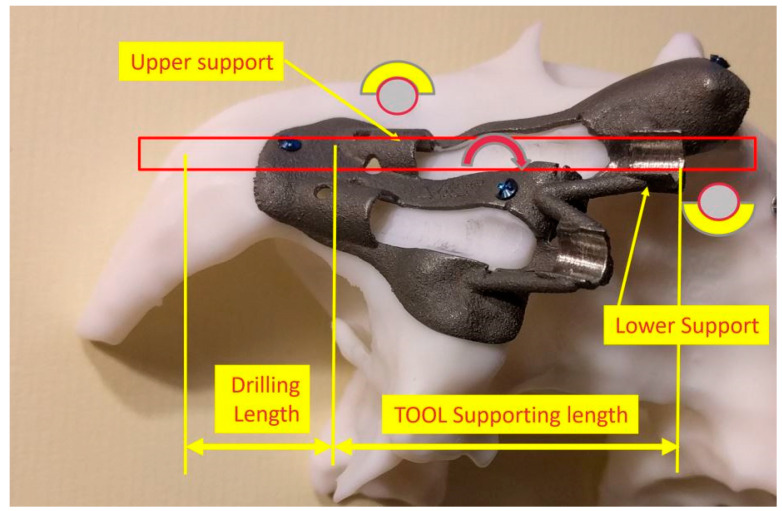
The 3D printed titanium surgical guide provides the support for the burs. Attention is paid to the supports which are designed in two parts, as upper and lower, in order to allow the surgeon to have a comfortable approach and to avoid a bur’s stop.

**Figure 7 ijerph-18-06142-f007:**
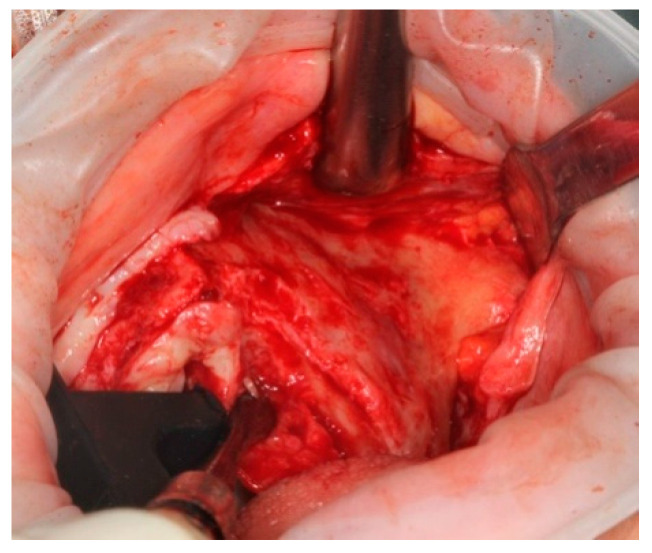
Palatal incision allowed a wide surgical access.

**Figure 8 ijerph-18-06142-f008:**
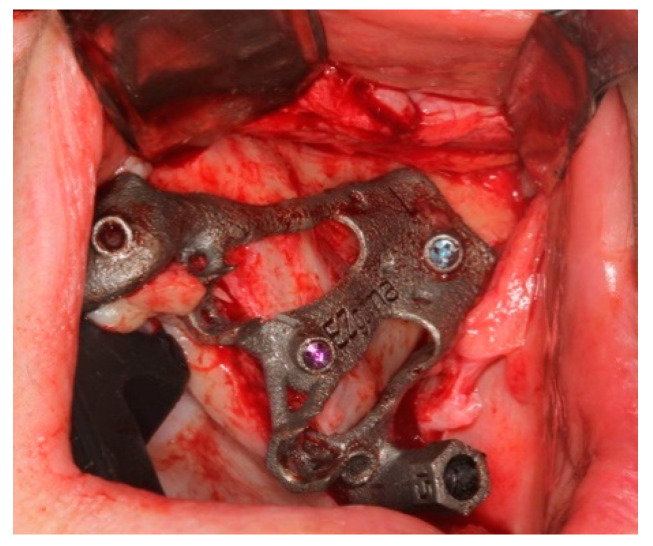
The guide perfectly fitted on the maxillary bone with 3 screws.

**Figure 9 ijerph-18-06142-f009:**
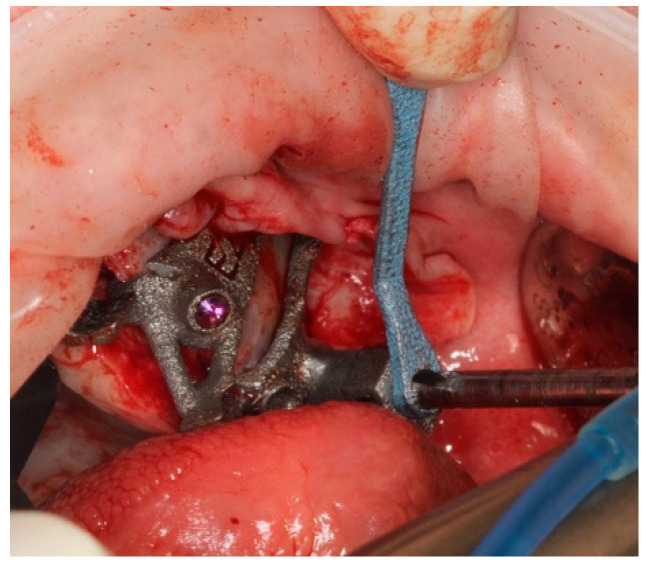
The first step was the pterygoid implant preparation, performed with the aid of a reduction spoon and a calibrated drill.

**Figure 10 ijerph-18-06142-f010:**
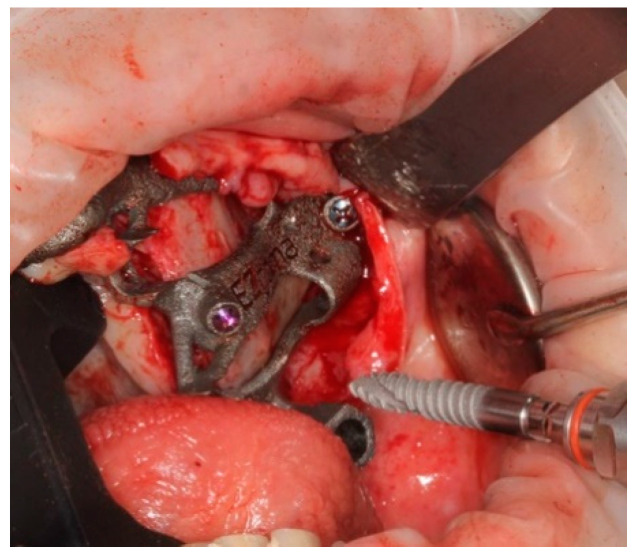
The placement of the pterygoid implant increased the guide’s stability for the following zygomatic implant site preparations.

**Figure 11 ijerph-18-06142-f011:**
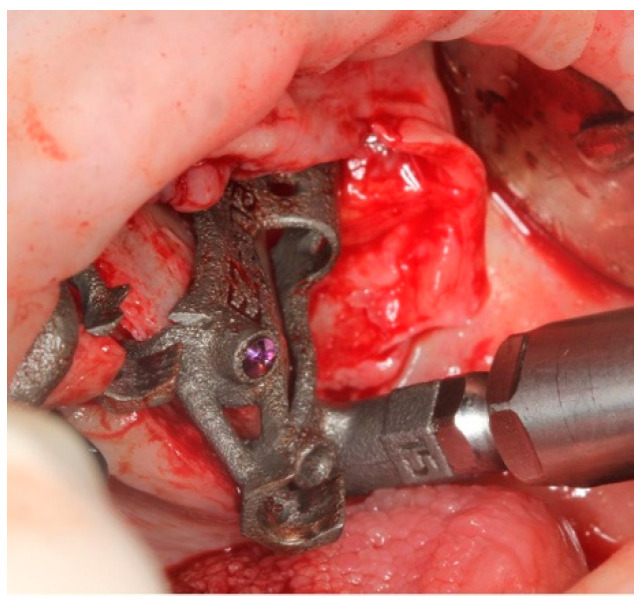
Pterygoid implant was placed by an implant mount that allow to position the implant as planned.

**Figure 12 ijerph-18-06142-f012:**
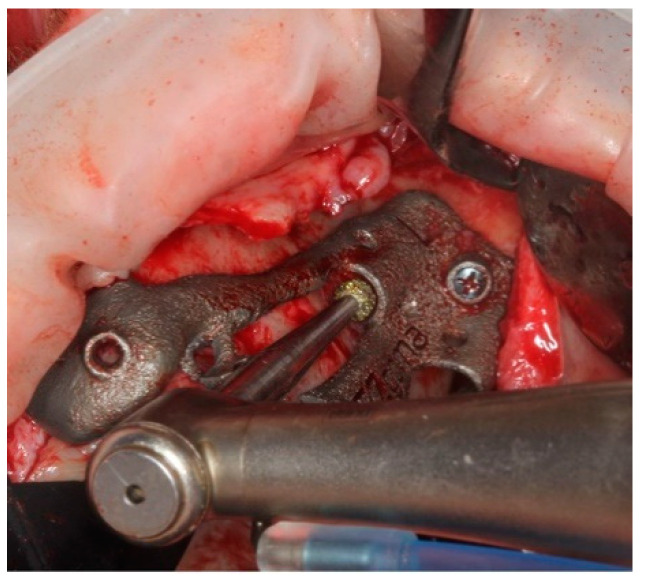
Round diamond bur was used to perform the primary corticotomy.

**Figure 13 ijerph-18-06142-f013:**
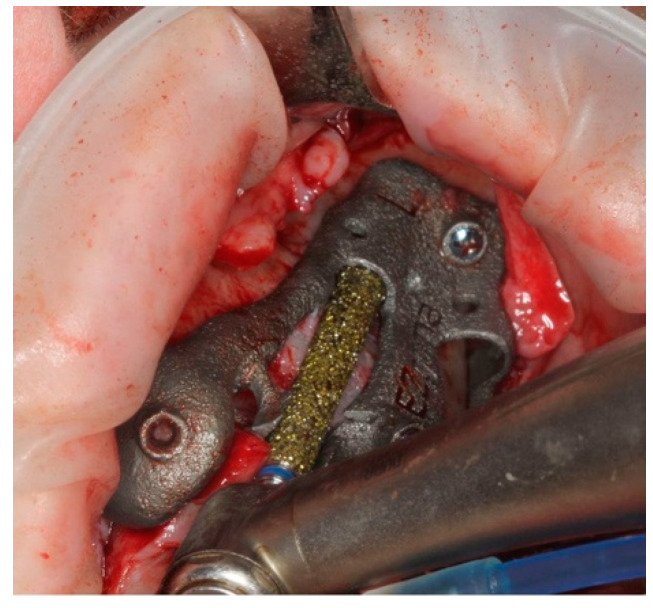
The outer wall of the sinus was prepared with cylindric diamond burs, in order to prepare the bone slot until it was adapted to the upper and lower support.

**Figure 14 ijerph-18-06142-f014:**
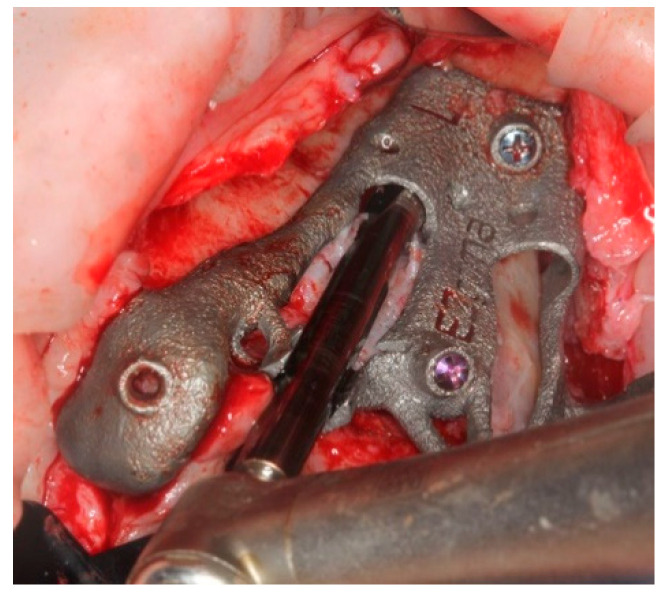
Centering spoon’s site preparation.

**Figure 15 ijerph-18-06142-f015:**
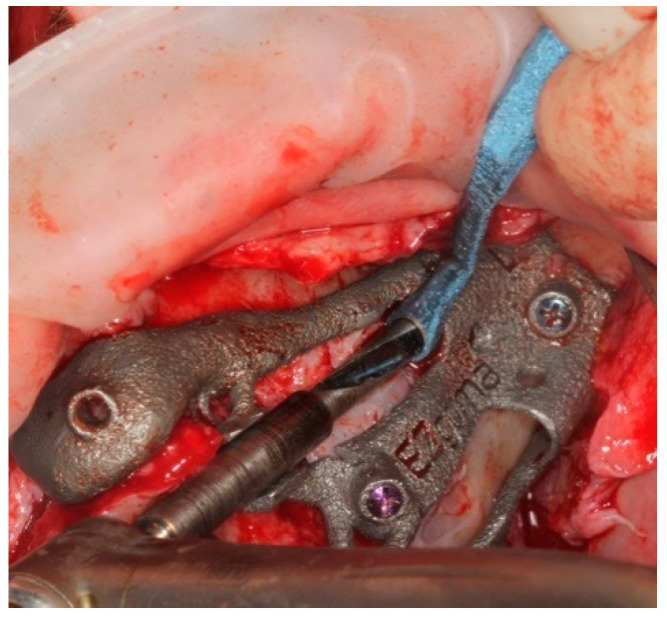
Main implant site preparation was carried out with a deepness bur and a reduction spoon.

**Figure 16 ijerph-18-06142-f016:**
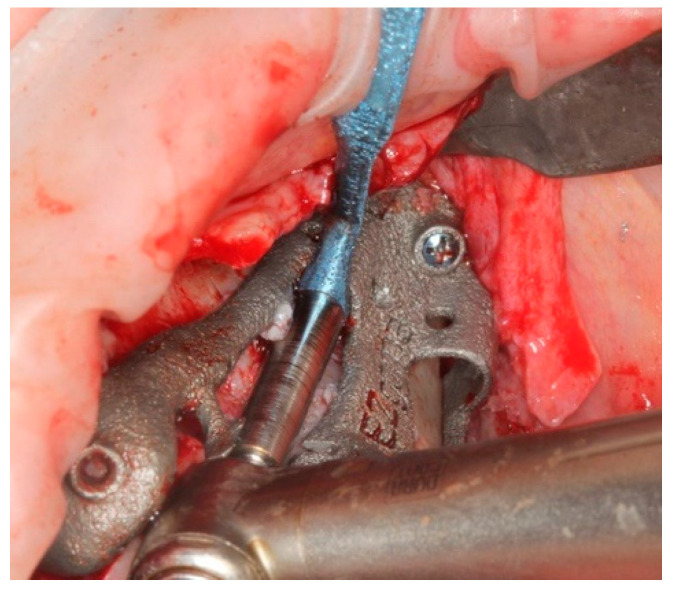
The deepness bur at the end of its path.

**Figure 17 ijerph-18-06142-f017:**
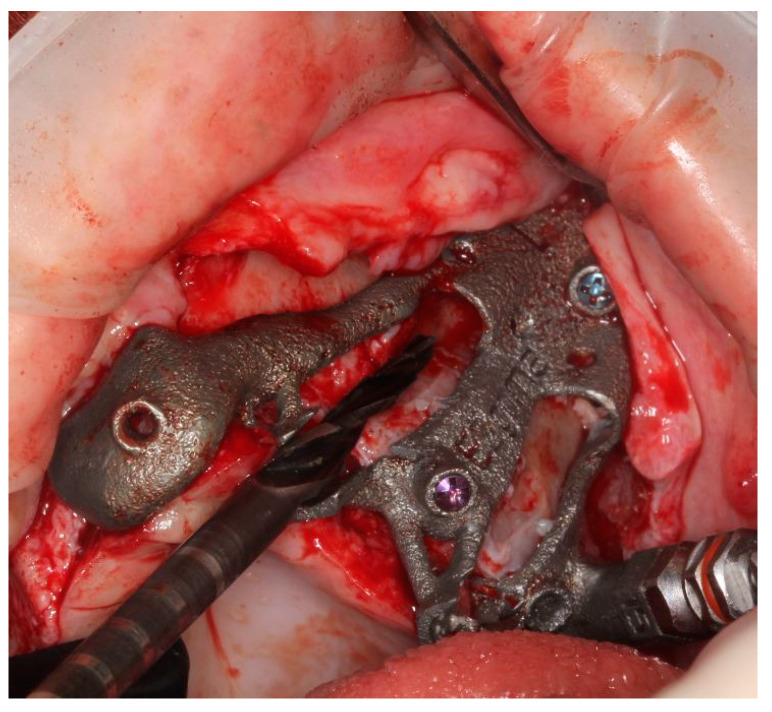
Last bur to finalize the implant site. Note that at this point no other tool, including reduction spoon is needed.

**Figure 18 ijerph-18-06142-f018:**
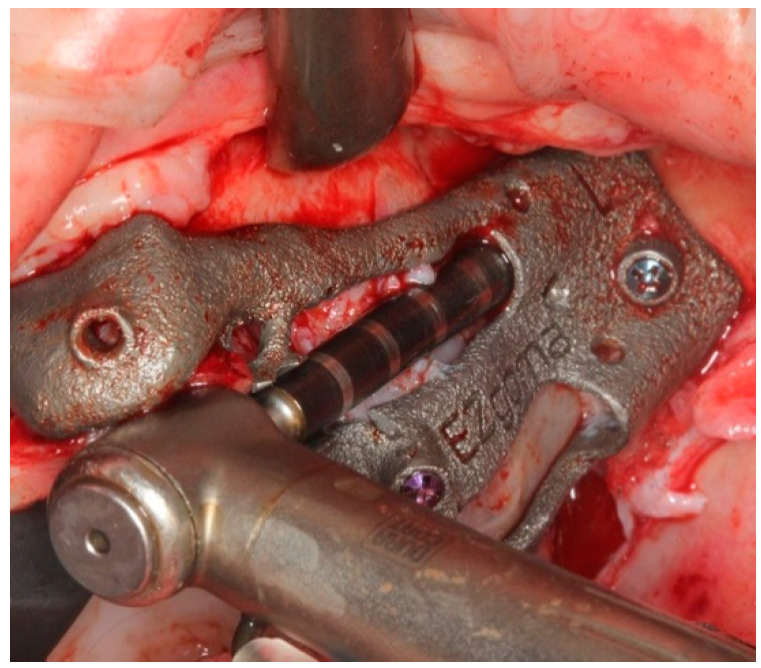
The final drill at the end of the preparation.

**Figure 19 ijerph-18-06142-f019:**
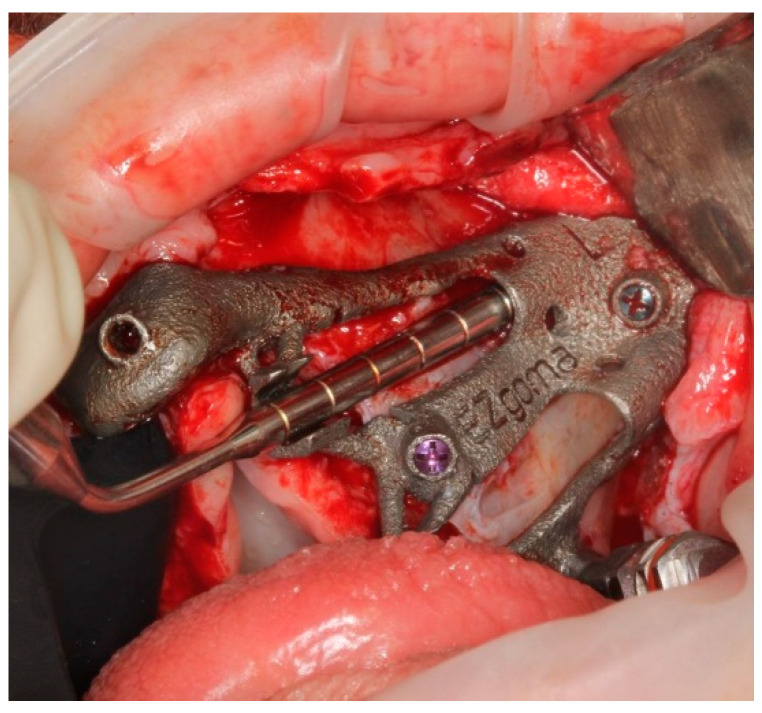
Depth caliper was used to assess the bone site preparation.

**Figure 20 ijerph-18-06142-f020:**
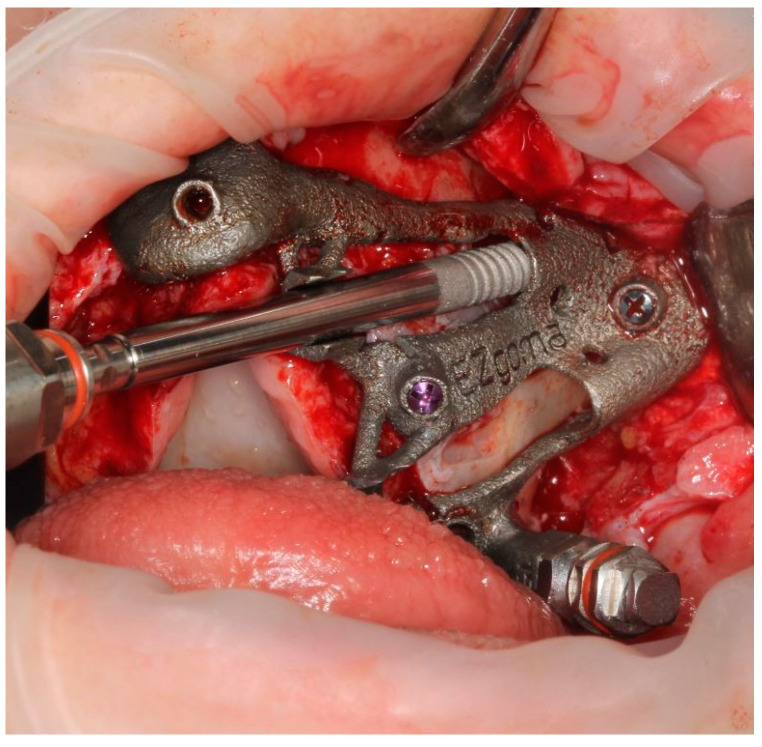
The zygomatic implant was screwed by a dedicated mounter to prevent implant’s deviation.

**Figure 21 ijerph-18-06142-f021:**
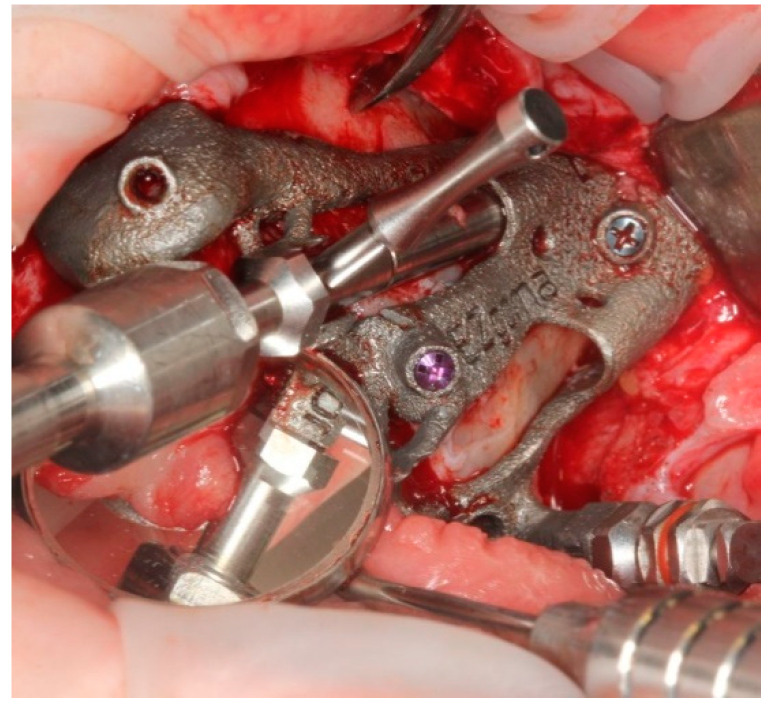
Implant placement was finalized with the aid of a pin to check even if the planned prosthetic connection orientation was achieved.

**Figure 22 ijerph-18-06142-f022:**
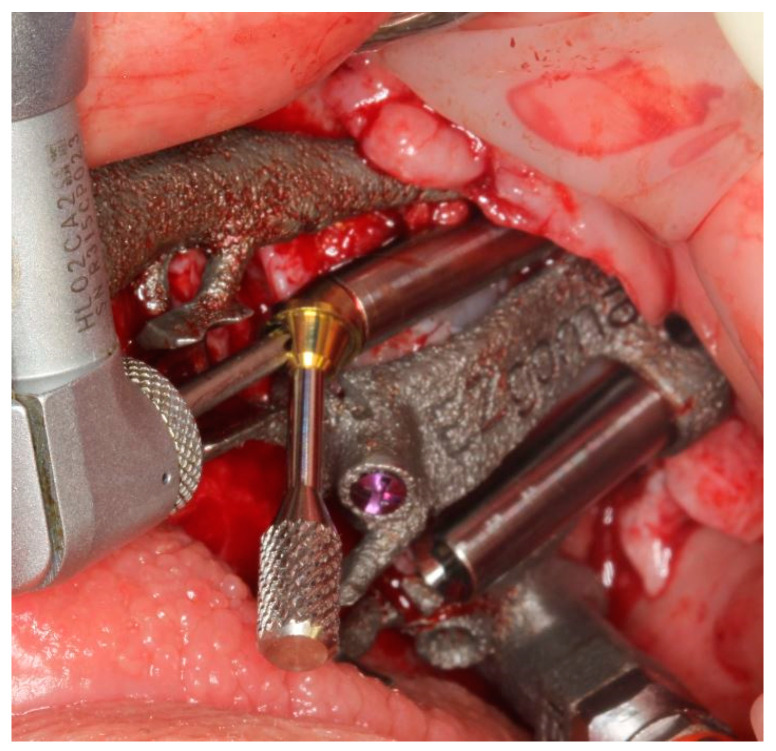
Multi-unit-abutment was screwed on the fixture with the surgical guide still in place.

**Figure 23 ijerph-18-06142-f023:**
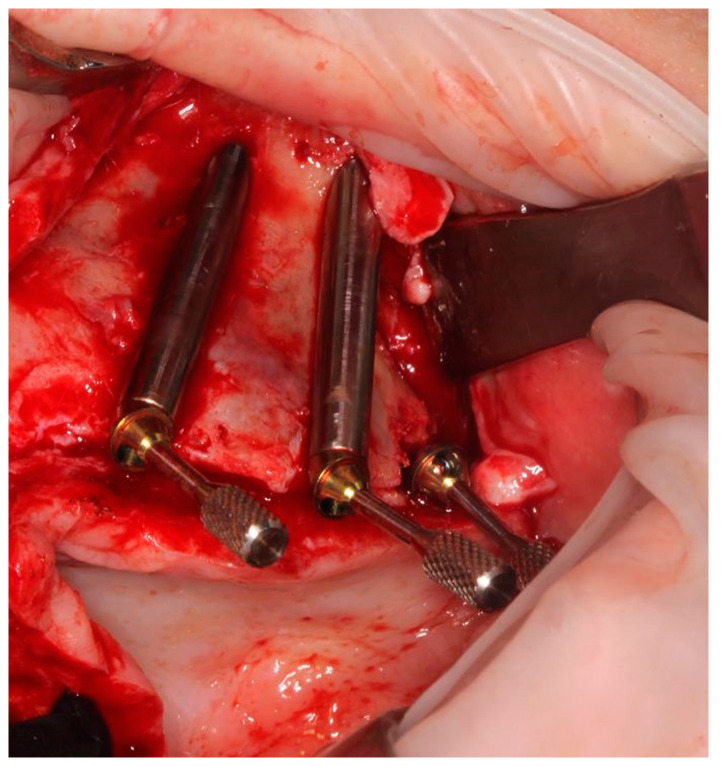
Implants and abutments in place confirming the surgical plan.

**Figure 24 ijerph-18-06142-f024:**
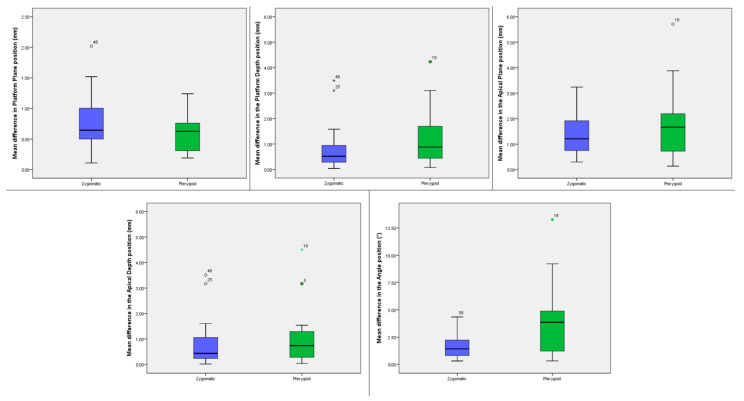
Box plots showing the median, quartile, minimum and maximum values of the mean differences in the platform plane (mm), platform depth (mm), apical plane (mm), apical depth (mm) and angle (°) between the zygomatic and the pterygoid implants compared to the virtual planning. Boxes contain 50% of all values; the horizontal lines inside the boxes indicate the medians and the vertical lines extend to 1.5 of the interquartile range.

**Table 1 ijerph-18-06142-t001:** Mean difference in the platform plane, platform depth, apical plane, apical depth and angle position between zygomatic and pterygoid implants.

	Zygomatic	Pterygoid	*p*-Value
Platform plane (mm)	0.76 ± 0.41	0.61 ± 0.28	0.144
Platform depth(mm)	0.93 ± 1.23	1.35 ± 1.45	0.256
Apical plane (mm)	1.35 ± 0.78	1.81 ± 1.47	0.213
Apical depth (mm)	1.07 ± 1.47	1.22 ± 1.44	0.711
Angle position (°)	1.69 ± 1.12	4.15 ± 3.53	0.006

**Table 2 ijerph-18-06142-t002:** Mean difference in the platform plane, platform depth, apical plane, apical depth and angle position between left and right side in both zygomatic and pterygoid implants.

Zygomatic			
	Left Side	Right Side	*p*-Value
Platform plane (mm)	0.86 ± 0.44	0.67 ± 0.36	0.163
Platform depth (mm)	0.64 ± 0.43	1.17 ± 1.59	0.204
Apical plane (mm)	1.32 ± 0.84	1.38 ± 0.76	0.799
Apical depth (mm)	0.65 ± 0.45	1.41 ± 1.88	0.095
Angle position (°)	1.65 ± 1.20	1.72 ± 1.09	0.864
**Pterygoid**			
Platform plane (mm)	0.62 ± 0.15	0.59 ± 0.37	0.801
Platform depth (mm)	1.24 ± 1.64	1.47 ± 1.30	0.741
Apical plane (mm)	1.70 ± 1.57	1.91 ± 1.43	0.762
Apical depth (mm)	1.09 ± 1.52	1.35 ± 1.43	0.703
Angle position (°)	3.89 ± 3.73	4.42 ± 3.49	0.748

## Data Availability

The data presented in this study are available on request from the corresponding author.

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
