# Peer review of "A Novel Guided Zygomatic and Pterygoid Implant Surgery System: A Human Cadaver Study on Accuracy"

_ijerph, 2021, doi:10.3390/ijerph18116142_

Round 1

Reviewer 1 Report

The manuscript reported a guided zygomatic and pterygoid implant surgery system. The authors should address my two main concerns listed below.

Please complete the Institutional Review Board Statement or indicate it in the Methods section whenever appropriate. Please state the number of human cadavers involved.

The CT scan parameters should be reported, especially the voxel size. This might affect the linear/angular measurement limit. How could the authors measure so precisely with 3 decimal places? A normal CBCT or CT scan has a voxel size of as fine as 0.08 mm. If the superimposed pre-/post- implant images had one voxel deviation in position, the difference would be 0.08 mm (2 decimal places only). Please elaborate.

Author Response

The manuscript reported a guided zygomatic and pterygoid implant surgery system. The authors should address my two main concerns listed below.

Please complete the Institutional Review Board Statement or indicate it in the Methods section whenever appropriate. Please state the number of human cadavers involved.

ANSWER: The cadavers were donated by individuals for their use in scientific purposes and an official laboratory permission to work on cadavers was obtained from Italian competent authority (Prot. Nr 08 del 05 Maggio 2021).The permission from the Teaching and Research Center Foundation is attached to this submission of the article as a separate pdf file.

LINE 109: The following paragraph was added.  Common rules/guidelines applied in European Union which was used in this study while working on cadavers were as follows:

  • Cadavers were treated with respect at all times
  • A professional attitude was applied during all lab
  • Human cadaver material was not removed from the laboratory under any circumstances.
  • No photographs or video cameras were used in the laboratory
  • Only health professionals enrolled in the course and instructors entered the lab
  • All cadaver material remained at the assigned dissection table
  • Incomplete dissections or intentional destruction of dissected structures was considered unprofessional behavior and work area was kept as clean as possible.

The CT scan parameters should be reported, especially the voxel size. This might affect the linear/angular measurement limit. How could the authors measure so precisely with 3 decimal places? A normal CBCT or CT scan has a voxel size of as fine as 0.08 mm. If the superimposed pre-/post- implant images had one voxel deviation in position, the difference would be 0.08 mm (2 decimal places only). Please elaborate.

ANSWER: Dear doctor, thank you for your valuable contribution for improving our work. As you suggested we agree with your suggestion, we added CT parameters and we decided to change the results with 2 decimals places instead of 3 decimals ones.

Reviewer 2 Report

The authors Grecchi et al., represented an excellent research article entitled "A novel guided zygomatic and pterygoid implant surgery system: A human cadaver study on accuracy". The research article is extremely helpful for surgical practice purpose as this study uses human cadaver model. There are some minor issue to be resolved before acceptance. I recommend for minor revision. 

  1. Please discuss the novelty of this study compared to already reported articles. Please enrich the Introduction section with similar work reference.
  2. What is the post surgical observation process please mention. How long this post surgical observation is acceptable as the study models are human cadaver. 
  3. What are the additional important guidelines to be followed to study human cadaver model please incorporate in tabular form or in notes. 
  4. Is there any additional ethical issues or guidelines to be followed? As the authors are from two different countries what are the common rules they need to follow. Though the rules may vary with continent. But authors may interested to know any common laws or rules. 

Author Response

The authors Grecchi et al., represented an excellent research article entitled "A novel guided zygomatic and pterygoid implant surgery system: A human cadaver study on accuracy". The research article is extremely helpful for surgical practice purpose as this study uses human cadaver model. There are some minor issue to be resolved before acceptance. I recommend for minor revision. 

ANSWER: Dear doctor, thank you for your appreciation of our study. We believe that your valuable comments shall improve our work.

  1. Please discuss the novelty of this study compared to already reported articles. Please enrich the Introduction section with similar work reference.

ANSWER: Further discussions have been added to the introduction. We were unable to find ex vivo or in vivo clinical studies investigating accuracy and performing a completely guided surgery. There is a shortage of such articles in the literature. That was the reason in first place this part might seems lacking.

  1. What is the post surgical observation process please mention. How long this post surgical observation is acceptable as the study models are human cadaver. 

ANSWER: After the CT scan, the post surgical observation process included the research of any kind of perforation of the orbital cavity and the fracture of zygomatic bone.

  1. What are the additional important guidelines to be followed to study human cadaver model please incorporate in tabular form or in notes. 

ANSWER: The cadavers were donated by individuals for their use in scientific purposes and an official laboratory permission to work on cadavers was obtained from Italian competent authority.  Common rules/guidelines applied in European Union which was used in this study while working on cadavers were as follows:

  • Cadavers were treated with respect at all times
  • A professional attitude was applied during all lab
  • Human cadaver material was not removed from the laboratory under any circumstances.
  • No photographs or video cameras were used in the laboratory
  • Only health professionals enrolled in the course and instructors entered the lab
  • All cadaver material remained at the assigned dissection table
  • Incomplete dissections or intentional destruction of dissected structures was considered unprofessional behavior and work area was kept as clean as possible.

These rules were added to the text:

Common rules/guidelines applied while working on cadavers were as follows:

  • Cadavers were treated with respect at all times
  • A professional attitude was applied during all lab
  • Human cadaver material was not removed from the laboratory under any circumstances.
  • No photographs or video cameras were used in the laboratory
  • Only health professionals enrolled in the course and instructors entered the lab
  • All cadaver material remained at the assigned dissection table
  • Incomplete dissections or intentional destruction of dissected structures was considered unprofessional behavior and work area was kept as clean as possible.

  1. Is there any additional ethical issues or guidelines to be followed? As the authors are from two different countries what are the common rules they need to follow. Though the rules may vary with continent. But authors may interested to know any common laws or rules. 

ANSWER: The authors are from Italy and Israel. The study was conducted in Italy and because of this EU rules were applied while working on cadavers. Common rules/guidelines applied while working on cadavers are listed above and in Line 114 and followign in the manuscript.

Reviewer 3 Report

Dear author I think technology always helps and will help clinicians and this techique and this article could create a new protocol for zygomatic implants.

I agree with the methodology and the conclusion of this article.

You should check english for few typing errors and you should amplify the discussion. It is to short compared to results and introduction.

You should also provide other background and include other relevant references in introduction and discussion.

First of all, I think you should clarify the ethical Review Board Statement because the study is on human cadavers, it means human samples.

LINE 76-78. You state that there are few studies regarding how surgical experience can affect the success of dental implants. Between the studies that you cite in this sentece you should add the following one on this topic.

Barone, A.; Toti, P.; Marconcini, S.; Derchi, G.; Saverio, M.; Covani, U. Esthetic Outcome of Implants Placed in Fresh Extraction Sockets by Clinicians with or without Experience: A Medium-Term Retrospective Evaluation. Int. J. Oral Maxillofac. Implants 201631, 1397–1406.

LINE 348-349. You correctly state that in order to test implant deviations and to plan the zygomatic implant position you should use a guide and most of all a CT-scan to have a 3-dimentional immagine. Anyway you should also also admitt in the discussion that for the implant maintainment and check-up you have to use endoral radiography so it could be usefull to standardize the endoral radiography to have a two-dimensional immagine of zygomatic implant and to evaluate the marginal bone level every year. In this case you should cite the following 2 articles to try to standardize the implant position evaluated by b-dimensional radiography:

De Santis, D.; Graziani, P.; Castellani, R.; Zanotti, G.; Gelpi, F.; Marconcini, S.; Bertossi, D.; Nocini, P.F. A New Radiologic Protocol and a New Occlusal Radiographic Index for Computer-Guided Implant Surgery. J. Craniofac. Surg. 201627, e506–e510.

Cosola, S.; Toti, P.; Peñarrocha-Diago, M.; Covani, U.; Brevi, B.C.; Peñarrocha-Oltra, D. Standardization of three-dimensional pose of cylindrical implants from intraoral radiographs: A preliminary study. BMC Oral Health. 202121, 100.

Author Response

Dear author I think technology always helps and will help clinicians and this techique and this article could create a new protocol for zygomatic implants.

I agree with the methodology and the conclusion of this article.

ANSWER: Dear doctor, thank you for your appreciation and valuable contribution as comments that can improve our work.

You should check english for few typing errors and you should amplify the discussion. It is to short compared to results and introduction.

You should also provide other background and include other relevant references in introduction and discussion.

ANSWER: English was re-checked; discussion and introduction were enriched with additional references.

First of all, I think you should clarify the ethical Review Board Statement because the study is on human cadavers, it means human samples.

ANSWER: The cadavers were donated by individuals for their use in scientific purposes and an official laboratory permission to work on cadavers was obtained from Italian competent authority.  Common rules/guidelines applied in European Union which was used in this study while working on cadavers.

LINE 76-78. You state that there are few studies regarding how surgical experience can affect the success of dental implants. Between the studies that you cite in this sentece you should add the following one on this topic.

ANSWER: The following article was added to the introduction section at line 59.

Barone, A.; Toti, P.; Marconcini, S.; Derchi, G.; Saverio, M.; Covani, U. Esthetic Outcome of Implants Placed in Fresh Extraction Sockets by Clinicians with or without Experience: A Medium-Term Retrospective Evaluation. Int. J. Oral Maxillofac. Implants 2016, 31, 1397–1406.

LINE 348-349. You correctly state that in order to test implant deviations and to plan the zygomatic implant position you should use a guide and most of all a CT-scan to have a 3-dimentional immagine. Anyway you should also also admitt in the discussion that for the implant maintainment and check-up you have to use endoral radiography so it could be usefull to standardize the endoral radiography to have a two-dimensional immagine of zygomatic implant and to evaluate the marginal bone level every year. In this case you should cite the following 2 articles to try to standardize the implant position evaluated by b-dimensional radiography:

De Santis, D.; Graziani, P.; Castellani, R.; Zanotti, G.; Gelpi, F.; Marconcini, S.; Bertossi, D.; Nocini, P.F. A New Radiologic Protocol and a New Occlusal Radiographic Index for Computer-Guided Implant Surgery. J. Craniofac. Surg. 2016, 27, e506–e510.

Cosola, S.; Toti, P.; Peñarrocha-Diago, M.; Covani, U.; Brevi, B.C.; Peñarrocha-Oltra, D. Standardization of three-dimensional pose of cylindrical implants from intraoral radiographs: A preliminary study. BMC Oral Health. 2021, 21, 100.

ANSWER: The articles listed above were added to discussions section in a separate paragraph starting at line 411:

This present cadaver study was performed in an anatomical laboratory environment, however in real life clinicians perform zygomatic implant surgery on patients with extremely atrophic maxillary bone. Management of oral rehabilitation in such patients can be quite demanding, and one of the key factors is the careful follow-up period. The marginal bone resorption and changes in bone levels must be evaluated at least every two years. Especially, for the specific extra-sinus placement that is typical of the presented surgical protocol, peri-implant mucosal situation must be additionally controlled. For this purpose, it is mandatory to annually remove the prostheses to check the oral hygiene status of prosthesis and the status of the abutments that are placed over the zygomatic implants. The radiographic assessment of the on-going peri-implant vertical bone loss after implant placement is considered as an essential issue for clinicians. Cosola et al., proposed a method of standardization of two-dimensional radiographs that can allow the clinicians to minimize the patient's exposure to ionizing radiations for the measurement of marginal bone levels around dental implants [45]. Such methods can be critical in order to evaluate the bone changes around the zygomatic implants at the follow-up period and have a great impact on the long-term successful results.

At line 429: In cases of guided implant surgery, patients with limited arch space can be a challenging situation specially for zygomatic implant insertions. In such cases, various protocols were introduced in literature as a solution. De Santis et al., in a clinical study evaluated a novel radiologic protocol and a new occlusal radiographic index that can give the clinician the possibility of identifying patients with limited inter-arch space. As a result, the new radiological occlusal index made with condensation silicone (Sandwich Index) proved to be effective in reproducing the maxillary forced maximum opening position during the initial planning phase. Additionally, their method prevented errors in the inclusion or exclusion of patients suitable for NobelGuide treatment [46]. The EZgoma guide system represented in this study, is a suitable method even for patients with limited mouth opening. This guide system is fixed unilaterally, which is easy to use.

Round 2

Reviewer 1 Report

The authors have addressed my concerns.